# Experimental Study on Mechanical Properties of Hemp Fibers Influenced by Various Parameters

**João Ribeiro** [1,2] , **Guilherme Bueno** [1], **Manuel Rodríguez Martín** [3,*] and **João Rocha** [1]

1 Laboratório Associado para a Sustentabilidade e Tecnologia em Regiões de Montanha (SusTEC), Instituto Politécnico de Bragança, Campus Santa Apolónia, 5300-253 Bragança, Portugal; jribeir@ipb.pt (J.R.); irigoyenguilherme@gmail.com (G.B.); jrocha@ipb.pt (J.R.)
2 CIMO—Mountain Research Center, Campus Santa Apolónia, 5300-253 Bragança, Portugal
3 Department of Mechanical Engineering, Universidad de Salamanca, Av. Requejo 33, 49029 Zamora, Spain
* Correspondence: ingmanuel@usal.es

**Abstract:** Hemp fibers produced by different extraction techniques that were cultivated in the Portuguese districts of Bragança and Mirandela under various nitrogen fertilization and planting density conditions are examined and mechanically described in this paper. The objective of this study is to evaluate the influence of sowing density, nitrogen fertilization, sample location, and extraction method in order to improve the mechanical properties of hemp fibers. To achieve this, we determined the modulus of elasticity, the tensile strength, and the density. The mean value for the modulus of elasticity was $92.44 \pm 7.44$ GPa, the mean of tensile strength was $564.98 \pm 167.03$ MPa, and the mean of the density was $1.64 \pm 0.24$ g cm$^{-3}$. We performed a statistical analysis of all parameters using ANOVA and found that the retting method had the greatest influence among all parameters. The associated effects of nitrogen fertilization and sowing density revealed an important influence on tensile strength and specific tensile strength, respectively.

**Keywords:** natural fibers; sowing density; hemp fiber; nitrogen fertilization; mechanical properties





## 1. Introduction

Natural fibers are a viable substitute for the conventional synthetic fibers employed for reinforcing composite materials, offering a diminished ecological footprint [1]. Additionally, natural fibers offer the advantages of recyclability, renewability, low density, abundant availability, and cost-effectiveness compared with synthetic fibers [2]. This kind of fiber is usually associated with a resin obtaining a composite [3], which can be used in many applications [4]. Associated with natural fibers, the use of biodegradable resins has also been studied in recent years, namely, PLA (polylactic acid) [5] and PDMS (polydimethyl-siloxane) [6,7], among others. In order to obtain a good bond between natural fibers and resins, there are new technologies that range from surface modification of fibers [8,9] to the use of nanoparticles [10,11]. Once hemp fiber gained use in many applications, the automotive industry led the way in adopting it in their composites materials, for example, hemp fiber-epoxy is particularly attractive to vehicle applications [5].

Hemp's abundance of beneficial agronomic, ecological, and pharmaceutical properties qualifies it as a valuable raw material for a diversity of conventional (medicine, food, fiber, oil) and advanced industrial products [12]. Hemp has the potential to be a sustainable and ecologically beneficial crop. Because of its fast growth and productivity, it is one of the most efficient $CO_2$ biomass converters. Hemp has proven to be an exceptional carbon trap, absorbing more $CO_2$ per hectare than most agricultural commodity crops and even forests [13]. Hemp fiber is made up of three main parts—the lumen, the secondary wall, and the primary external wall (Figure 1). The walls consist of layers of fibrils that are linked together by lignin. The secondary wall is made up of microfibrils of cellulose arranged in spirals and linked to hemicelluloses [6].

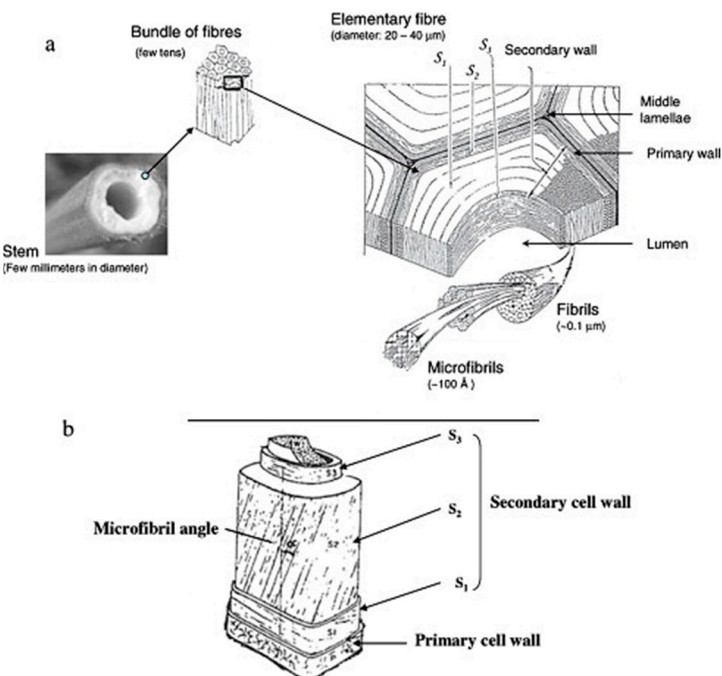

**Figure 1.** Hemp fiber structure. (**a**) Schematic representation of a natural fibre from stem to microfibrils (inspired from Frey-Wyssling). (**b**) Schematic structure of a natural fibre cell wall. Reprinted/adapted with permission from Ref. [14]. 2023, Vincent Placet.

According to Ref. [7], hemp fiber properties depend on the growing conditions, retting method, and treatment; thus, it is important to know the influence of these parameters on the mechanical properties of the fiber. The information available on the effect of planting density and nitrogen fertilization is limited. The optimization of planting density and nitrogen fertilization to improve the mechanical properties have not been studied for hemp fibers.

Young's modulus, tensile strength, and density for hemp fiber are well known in the literature, varying from 58 GPa to 70 GPa [7–10], 550 MPa to 1110 MPa [7,9–11], and 1.4 g cm$^{-3}$ to 1.6 g cm$^{-3}$ [1,7,9,10], respectively. The elongation at break point was studied, and its values in the literature are 1.6% [7,15].

Information about the retting method suggests that the removal of lignin, pectin, and hemicellulose provided by NaOH improves the mechanical properties due to the better packing and orientation of cellulose chains, though over-treatment reduces fiber properties [16,17].

Studies investigating the effect of nitrogen fertilization and planting density on hemp cultivation were conducted with the same samples used for this research. The study showed that higher planting density leads to higher fiber production and that increasing nitrogen fertilization provides less fiber production [18]. Other studies on the effect of planting density and nitrogen fertilization include one by Tang et al. [19], showing a direct relation between planting density and nitrogen fertilization on stem yield. Increasing planting density and nitrogen fertilization above 120 plants m$^{-2}$ and 60 kg ha$^{-1}$ did not result in a significant increase in stem yield.

The contribution of this work is to analyze bundles of hemp fibers grown in regions of Mirandela and Bragança, located in northeastern Portugal, for distinct conditions of fertilization with nitrogen and cultivation density. The main goal of this study is to determine the effects of fertilization with nitrogen, sowing density, retting method, and sample position on hemp fibers' mechanical properties with different climatic conditions compared with those applied in other previous studies [16,17].

Since this work will improve the production of natural fibers and the manufacture using by-products, it allows manufacturing with natural materials, favoring the fight against

climate change (SDG-13), favoring the use of renewable resources (SDG-15), and allowing the manufacture of more sustainable products and boosting the circular economy (SGD-12).

## 2. Materials and Methods

This study used seeds of *Cannabis sativa* L., "Futura 75" variation obtained from "Coopérative des Producteurs de Semences de Chanvre". The crop was grown in two different regions of Portugal: Bragança and Mirandela. The regions were chosen based on their climactic differences; Bragança is in an area known locally as "cold land", where winter and summer temperatures are lower than in Mirandela, which is located in an area known locally as "warm land". Due to mechanical issues with the irrigation system, the samples of Mirandela were flooded with water a few days after sowing. Therefore, only one-third of the samples had appropriate seed germination. Due to this, more effort was given to the results collected on fibers from Bragança.

The sample setup is displayed below (Figure 2), where each row represents a different sowing density and each column a different nitrogen fertilization. Three different sowing densities were named D25, D50, and D100, with each one representing the number of seeds corresponding to 25, 50, and 100 kg ha$^{-1}$, respectively. For nitrogen applications, there were four treatments—N0, N50, N100, and N200—corresponding to no addition (N0) and the addition of 50, 100, and 200 kg ha$^{-1}$ of nitrogen, respectively. Half of the nitrogen was added when sowing and the other part was applied above ground four weeks later. All samples were given a base additional 100 kg ha$^{-1}$ of potassium and phosphorus.

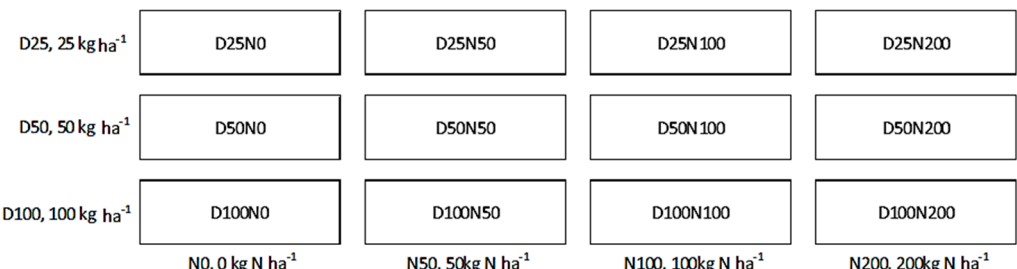

**Figure 2.** Experiment setup for sample collection.

External and internal samples were analyzed; external samples were located near the edge of the cultivation zone and internal samples were located near the center of the cultivation region.

The sowing took place at the end of May 2019 and the samples were grown together with Maize, with a total irrigation of around 450 mm during the trial. The time from sowing to harvest was roughly 90 days. Each plot was 1.5 × 4 m with walkways of 50 cm in both directions.

Two retting extraction methods were used: anaerobic retting and chemical retting by boiling in a NaOH solution. The first method was based on cutting a 400 mm sample from three average-size stems randomly chosen from each sample; the stems were submerged in water until the bark was easily removed from the woody core. The water used was changed three times throughout the process to reduce changes in pH and better simulate what is described in the literature. After removing the bark, the samples were dried at 70 °C.

Chemical retting was achieved by boiling the samples in a 0.35% NaOH solution for 1.5 h to loosen the bark from the core. The bark was boiled in a 2% NaOH solution; then, it was completely rinsed in tap water and dried at 70 °C.

To execute density measurements using a water pycnometer, it was necessary to calibrate it to determine its correct volume. There were measurements of the empty mass ($M_1$) and the mass of the pycnometer full of distilled water ($M_2$). Using Equation (1),

the correct volume ($V_{pycnometer}$) was calculated. This procedure was repeated five times (adopting the mean value).

$$V_{pycnometer} = \frac{(M_2 - M_1)}{\rho_{H2O}} \tag{1}$$

Using a pycnometer to estimate the correct density of material requires some calculations. First, it was necessary to measure the mass of the fibers ($M_3$). The procedure was performed using a scale Kern/ACJ/ACS 220-4, and a bunch of fibers was randomly selected and weighted. After that, the fibers were inserted into the pycnometer and then filled with distilled water. Finally, the fiber, water, and pycnometer all together ($M_4$) were weighted. The density of fiber ($\rho_{fiber}$) was calculated using Equation (2).

$$\rho_{fiber} = \frac{M_3}{\left( V_{pycnometer} - \frac{M_4 - M_3 - M_1}{\rho_{H2O}} \right)} \tag{2}$$

Tensile strength tests were performed using an adaptation of ASTM C1557-20 [20] to avoid fiber sliding. The fiber was mounted directly to the gripping system using two metal sheets covered with sandpaper for better gripping (Figure 3). After changing the gripping system, fiber sliding decreased and better results were obtained.

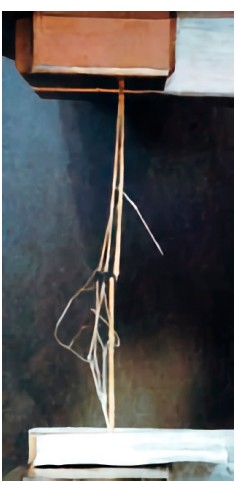

**Figure 3.** The gripping system used for tensile strength and Young's modulus test.

Tests were conducted using a universal test machine model Shimadzu Autograph AGS-X 10KN with a cross-head speed of 3 mm/min and a gauge length of 60 mm. Before the test started, a pre-load of 1 N was made for all specimens. The specimens were selected for better width and thickness uniformity. Three measurements of thickness and width were made to determine the area mean value.

Four samples of fibers with higher tensile strength were selected for Young's Modulus determination. Young's Modulus determination was achieved according to ASTM C1557 with three lengths of fibers being used: 40 mm, 50 mm, and 60 mm. The specimens of each plot were selected based on the uniformity of dimensions. System compliance was determined for the test machine, gripping system, and hemp fiber. Tensile tests were conducted for each plot; from these, force versus cross-head displacement curves were obtained—further, from these, system compliance and Youngs's Modulus curves were determined.

The ANOVA evaluation was used to investigate which parameters were more influential for tensile strength, specific tensile strength, and density of hemp fibers.

## 3. Results

### 3.1. Density

Table 1 shows density results as well as the standard deviation for all parameters: retting method, sample location, sowing density, and nitrogen fertilization. The average value for density in all samples was $1.65 \pm 0.25$ g cm$^{-3}$.

**Table 1.** Average density values and standard deviation for all analyzed parameters.

| | Density (g/cm$^3$) | Standard Deviation |
|---|---|---|
| Geographic location | | |
| Bragança | 1.66 | 0.29 |
| Mirandela | 1.63 | 0.25 |
| Retting Method | | |
| Water | 1.49 | 0.18 |
| NaOH | 1.81 | 0.26 |
| Samples location | | |
| External | 1.66 | 0.29 |
| Internal | 1.61 | 0.22 |
| Sowing Density | | |
| D25 | 1.58 | 0.31 |
| D50 | 1.79 | 0.32 |
| D100 | 1.66 | 0.23 |
| Nitrogen Fertilization | | |
| N0 | 1.58 | 0.22 |
| N50 | 1.58 | 0.27 |
| N100 | 1.68 | 0.27 |
| N200 | 1.77 | 0.31 |

### 3.2. Tensile Strength

The average value for tensile strength in all samples was $566.14 \pm 167.02$ MPa with a specific tensile strength of $353.19 \pm 114.02$ MPa/g cm$^{-3}$. Figure 4 shows the values of tensile strength with respect to the grouped data (nitrogen fertilization and sowing density).

Figure 5 shows the average values for specific tensile strength for the different nitrogen fertilization and sowing densities.

Table 2 shows the average values of tensile strength for all analyzed parameters as well as the standard deviation.

### 3.3. ANOVA Analysis

General linear model ANOVA evaluation was performed for all the parameters in Tables 3–5. Each row represents a determined parameter or an interaction between two distinct parameters, where DF is the degree of freedom, Adj. SS is the sum of squares, and Adj. MS are both the mean squares. The F-Ratio test is a statistical tool to verify which design parameters significantly affect the quality characteristic. This is defined as the ratio of the mean squared deviations to the mean squared error. The analysis allows us to estimate the contributions that report the percentage each source contributes to the total sequential sums of squares.

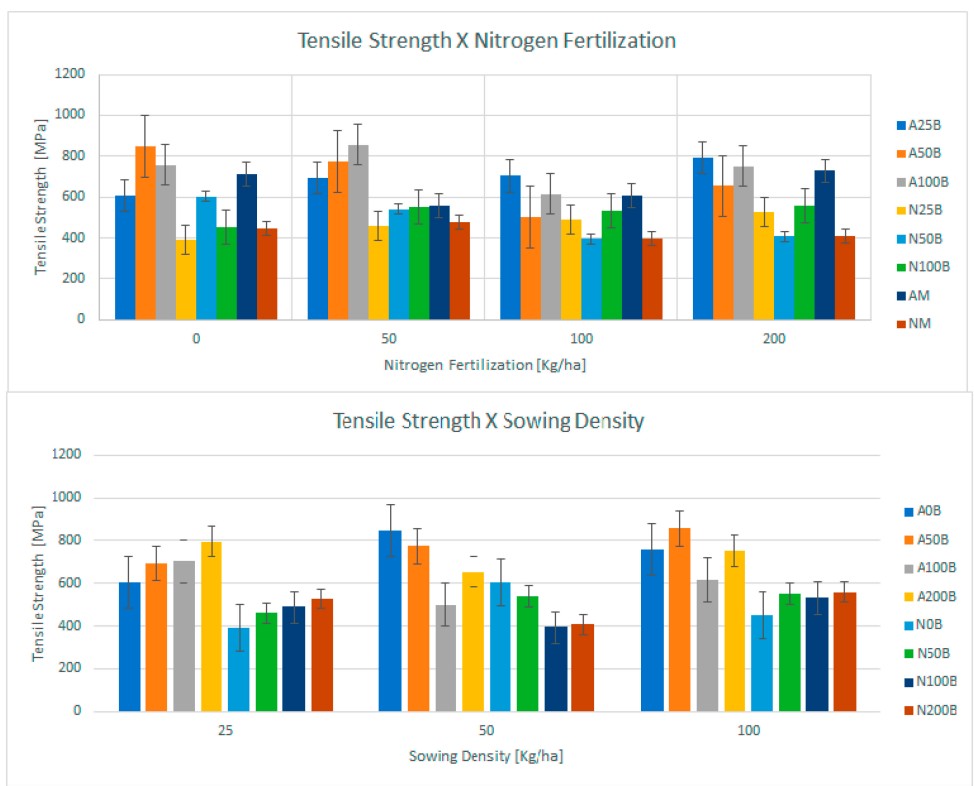

**Figure 4.** Tensile strength versus nitrogen fertilization and sowing density.

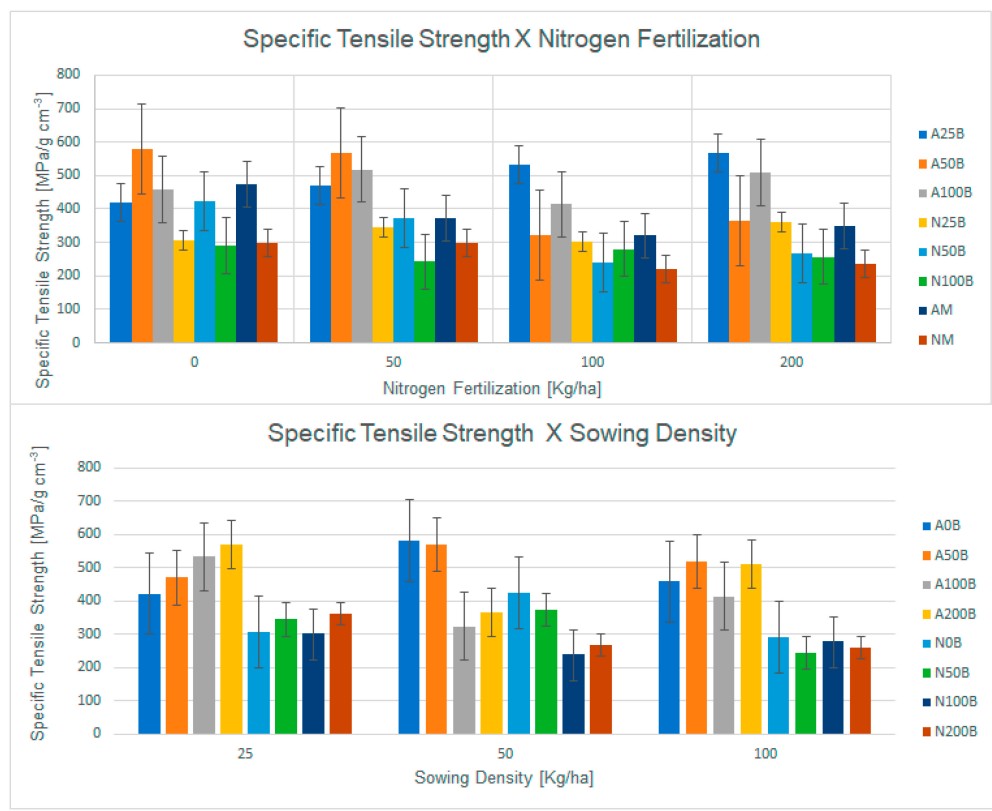

**Figure 5.** Specific tensile strength versus nitrogen fertilization and sowing density.

**Table 2.** Average tensile strength values and standard deviation for the analyzed parameters.

| | Tensile Strength (MPa) | Standard Deviation |
|---|---|---|
| Geographic location | | |
| Bragança | 596.19 | 164.98 |
| Mirandela | 493.36 | 150.44 |
| Retting Method | | |
| Water | 664.11 | 159.04 |
| NaOH | 466.56 | 105.33 |
| Samples location | | |
| External | 512.91 | 100.05 |
| Internal | 565.34 | 167.31 |
| Sowing Density | | |
| D25 | 583.02 | 138.96 |
| D50 | 589.88 | 162.54 |
| D100 | 634.34 | 138.47 |
| Nitrogen Fertilization | | |
| N0 | 608.62 | 173.33 |
| N50 | 645.51 | 153.01 |
| N100 | 540.10 | 107.09 |
| N200 | 615.43 | 145.90 |

**Table 3.** ANOVA evaluation for tensile strength measurements.

| Source | DF | Adj. SS | Adj. MS | F-Value | Contribution |
|---|---|---|---|---|---|
| Sowing Density | 2 | 12,421 | 6210.4 | 0.78 | 2.31% |
| Nitrogen | 3 | 35,693 | 11,897.5 | 1.5 | 6.64% |
| Retting | 1 | 82,696 | 82,695.7 | 10.43 | 15.38% |
| Place | 1 | 913 | 912.7 | 0.12 | 0.17% |
| Sow. Dens. × Nitr. | 6 | 102,798 | 17,133 | 2.16 | 19.12% |
| Sow. Dens. × Extr. | 2 | 603 | 301.4 | 0.04 | 0.11% |
| Nitr. × Extr. | 3 | 15,443 | 5147.8 | 0.65 | 2.87% |
| Extr. × Place | 1 | 22 | 21.8 | 0 | 0.00% |
| Error | 8 | 63,425 | 7928.2 | - | 11.80% |
| Total | 27 | 537,565 | - | - | - |

**Table 4.** ANOVA evaluation for specific tensile strength measurements.

| Source | DF | Adj. SS | Adj. MS | F-Value | Contribution |
|---|---|---|---|---|---|
| Sowing Density | 2 | 7034 | 3517 | 0.74 | 2.01% |
| Nitrogen | 3 | 18,698 | 6233 | 1.31 | 5.35% |
| Retting | 1 | 73,075 | 73,075 | 15.37 | 20.89% |
| Place | 1 | 15,495 | 15,495 | 3.26 | 4.43% |
| Sow. Dens. × Nitr. | 6 | 66,843 | 11,140 | 2.34 | 19.11% |
| Sow. Dens. × Extr. | 2 | 5427 | 2714 | 0.57 | 1.55% |
| Nitr. × Extr. | 3 | 3110 | 1037 | 0.22 | 0.89% |
| Extr. × Place | 1 | 2664 | 2664 | 0.56 | 0.76% |
| Error | 8 | 38,033 | 4754 | - | 10.87% |
| Total | 27 | 349,776 | - | - | - |

**Table 5.** ANOVA evaluation for density measurements.

| Source | DF | Adj. SS | Adj. MS | F-Value | Contribution |
|---|---|---|---|---|---|
| Sowing Density | 2 | 0.18390 | 0.09195 | 1.61 | 9.819% |
| Nitrogen | 3 | 0.17719 | 0.05906 | 1.04 | 9.461% |
| Retting | 1 | 0.00273 | 0.00273 | 0.05 | 0.146% |
| Place | 1 | 0.02170 | 0.02170 | 0.38 | 1.159% |
| Sow. Dens. × Nitr. | 6 | 0.10761 | 0.01794 | 0.31 | 5.746% |
| Sow. Dens. × Extr. | 2 | 0.10421 | 0.05211 | 0.91 | 5.564% |
| Nitr. × Extr. | 3 | 0.12083 | 0.04028 | 0.71 | 6.452% |
| Extr. × Place | 1 | 0.14787 | 0.14788 | 2.59 | 7.896% |
| Error | 8 | 0.45603 | 0.057004 | - | 24.351% |
| Total | 27 | 1.87276 | - | - | - |

In the above tables, Sow. Dens. × Nitr. means the interaction between sowing density and nitrogen fertilization in the soil; Sow. Dens. × Extr. means the interaction between sowing density and the retting method performed; Nitr. × Extr. means the interaction between the nitrogen fertilization and the retting method; finally, Extr. × Place means the interaction between the retting method and the geographic location.

As the reader can observe in Table 3, the parameter showing the highest contribution degree was the interaction "Sow. Dens. × Nitr" followed by retting method. On the other hand, the less influential parameters were the interactions "Ext. × Place" and "Sow. Dens. × Extr.".

In order to evaluate the contribution of each parameter, a multivariate analysis based on ANOVA was performed for specific tensile strength measurements. The software used was Minitab17. The parameters used and the interactions between them are represented as rows in Table 4.

As the reader can observe in Table 4, the parameter showing the highest contribution degree was "Retting", followed by "Sow. Dens. × Nitr.". The interactions of "Nitr. × Extr." and "Extr. × Place" did not show a significant influence.

ANOVA evaluation was also carried out for density measurements. The parameters used and the interaction between them are represented as the rows in Table 5. As the reader can observe in Table 5, the parameter showing the highest contribution degree was "Sow. Density" followed by "Nitrogen". However, the error value was significantly higher than for the rest of the evaluations.

### 3.4. Young's Modulus

Young's Modulus of four fibers with higher tensile strength was determined according to ASTM C1557. All the fibers with higher tensile strength were from Bragança and extracted by water. Table 6 shows the results from which we calculated that the average modulus of elasticity was $92.44 \pm 7.44$ GPa with an average specific Young's Modulus of $63.05 \pm 8.81$ GPa/g cm$^{-3}$ and an average failure strain of 1.81%.

**Table 6.** Young's Modulus results for the four higher tensile strength fibers.

| Sowing Density (kg/ha) | Nitrogen Fertilization (kg/ha) | Tensile Strength (MPa) | Young's Modulus (GPa) |
|---|---|---|---|
| 100 | 50 | 855.64 | 92.75 |
| 50 | 0 | 845.66 | 83.08 |
| 25 | 200 | 794.68 | 94.71 |
| 50 | 50 | 773.24 | 101.06 |

The average system compliance for the gripping system and the hemp fibers was 3.7 μm N$^{-1}$. Figure 6 shows the average stress versus strain for all four fiber samples analyzed.

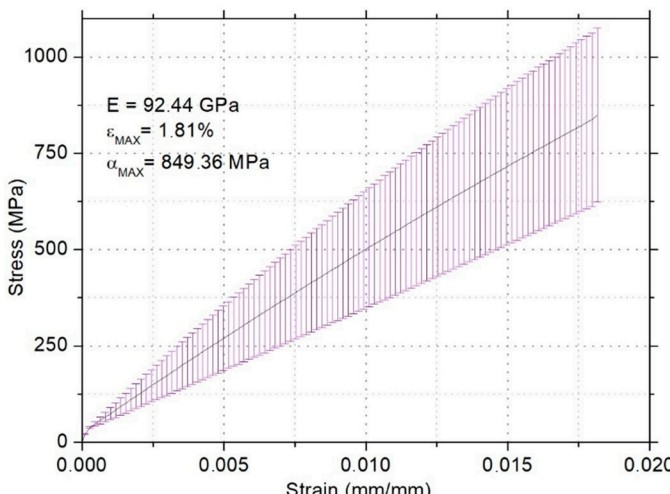

**Figure 6.** Stress versus strain curve for the average value of the four higher tensile strength samples.

## 4. Discussion

The mechanical properties of hemp fibers are dependent on many parameters such as chemical composition, defects, the presence/proportion of secondary fibers, and fiber diameter [21]. Fan et al. [22] verified that tensile strength decreases as fiber diameter increases, and this dependence of diameter is strictly related to both the number of single fibers contained in the fiber bundles and the number of fiber defects (i.e., dislocations). Nevertheless, these characteristics of the fibers (diameter, defects, and chemical composition, among others) are also dependent on plant sowing density and nitrogen fertilization [23]. In that study, the authors verified that high plant sowing density resulted in shorter plant height compared with low plant sowing density as the hemp plants tended to reach the reproductive stage early at high density. Stem diameter was inversely correlated with plant density. Reciprocally to seed production, inflorescence and stem yield were confirmed to increase in the early flowering as plant density increased. High N fertilization level had a positive impact on stems. Thus, as found by analyzing Table 2, the higher values will give thinner fibers; so, the value of tensile strength increases, as expected.

Retting is a biological process that removes non-cellulosic materials bound to the fiber bundle by enzymatic activities, resulting in detached cellulosic fibers. The water retting process develops anaerobic bacteria fermentation on fiber bundles to produce enzymes that hydrolyze fiber-binding components [24]. The main advantages of water retting are the production of retted fibers with high quality and great uniformity, despite the pollution problem originating from anaerobic bacterial fermentation. On the other hand, chemical retting allows the obtaining of clean and smooth surfaces in a short period of time; however, deterioration of the fibers' strength may occur if the NaOH concentration is greater than 1% [25]. For these reasons, it is possible to justify why the average value of tensile strength is higher for the fibers retted with water than those retted with chemical processes (NaOH), as shown in Table 2.

The average value of density is similar to that of other published works, as mentioned in [7]. The high value of standard deviation is related to the accuracy of water pycnometer tests, according to [26].

The ANOVA in Table 5 suggests that the influence of all parameters in density is lower than the error influence. The error influence is partly due to the lack of fit between samples. A moderate influence of sowing density (9.82%) and nitrogen fertilization (9.46%) is visible in the analyzed samples. Average values for all samples are equal when considering standard deviation. This indicates a low influence, direct or combined, in density measurements of the parameters analyzed in the present work.

Values for tensile strength and specific tensile strength are between the ones described in the literature in [7]. ANOVA suggests that the most influential factor on tensile strength

results is the combined effect of sowing density and nitrogen fertilization (19.12%) followed by retting method (15.38%); these results are limited to the samples from Bragança, considering the irrigation problems faced by the samples from Mirandela. The influence of sowing density and nitrogen fertilization on tensile strength values can be seen in Figure 4. There is a clear, linear trend line for sowing densities of 25 and 50 kg ha$^{-1}$.

ANOVA evaluation for specific tensile strength also suggests a higher influence of the retting method (20.89%) and a combined effect of sowing density and nitrogen fertilization (19.11%). A trend line can also be seen for sowing densities of 25 and 50 kg ha$^{-1}$ in Figure 5.

Young's modulus values are higher than those found in the literature, which, according to [7], are between 58 GPa and 70 GPa. Values for specific Young's modulus are also in disagreement with the literature, which, according to [7], should vary from 39 GPa/g cm$^{-3}$ to 47 GPa/g cm$^{-3}$. The discrepancy between the values found in this work and the ones from the literature is related to the high deviation between the experimental samples, according to [27]. Variations of mechanical properties and Young's modulus are related to fiber diameter, fiber length, and test speed. Tensile strength and elongation at break decreased while Young's modulus increased with fiber length. Increasing the test speed also increases the fibers' Young's modulus. Divergency can also be noticed in fail strength values, which, according to [7], are near 1.6%. Those divergences in results have been explained in terms of fiber structure, such as microfibrillar angle and cell structure [28].

In the present research work, the authors did not evaluate the influence of the irrigation process, despite the crop field being irrigated every fifteen days. However, Hackett [29] specified water stress as the greatest restricting factor influencing fiber yield and quality; for this reason, the authors carried out a short bibliographical research about the influence of the irrigation process specifically for growing hemp. Recently, Bajic et al. [30] studied the effect of irrigation on the yield and quality of hemp. They verified that irrigation did not influence the yield of hemp fiber in the first year of harvest and had a small influence in the second year of harvest. Another study was conducted by García-Tejero et al. [31] in 2014 to evaluate the impact of plant density and irrigation on the yield of hemp in a Mediterranean semi-arid region. They also verified that irrigation rates did not have a significant impact on the best results of yield. Considering these research works, for the present research, we believe that the irrigation did not have a crucial influence for the obtained results.

## 5. Conclusions

In this paper, the influence of different fiber parameters on mechanical properties was studied. For this purpose, an experimental study was carried out based on an experimental design that allowed us to obtain fibers with different characteristics and from different geographical areas of Portugal.

The optimal combination of sowing density, nitrogen fertilization, retting method, and place of growth to achieve a higher tensile strength was a sowing density of 100 kg ha$^{-1}$, nitrogen fertilization of 50 kg ha$^{-1}$, extracted with water, and cultivated in Bragança. The highest tensile strength was 855.65 MPa and the average tensile strength for samples from Bragança was 565.34 $\pm$ 166.97 MPa.

The ideal combination of sowing density, nitrogen fertilization, retting method, and place of growth was verified to be nitrogen fertilization of 100 kg ha$^{-1}$, sowing density of 25 kg ha$^{-1}$, retting with water, and cultivated in Bragança. The lowest density was 1.32 g cm$^{-3}$ and the average density for samples from Bragança was 1.65 $\pm$ 0.25 g cm$^{3}$.

For specific tensile strength, the optimal nitrogen fertilization, sowing density, retting method, and place of growing were obtained as sowing density of 50 kg ha$^{-1}$, nitrogen fertilization of 0 kg ha$^{-1}$, extracted with water, and cultivated in Bragança. The highest specific tensile strength was 580 MPa/g cm$^{-3}$ and the average value for samples from Bragança was 353.19 $\pm$ 114.02 MPa/g cm$^{-3}$.

The highest value for the modulus of elasticity was determined for fibers farmed in Bragança, with a nitrogen fertilization of 50 kg ha$^{-1}$, sowing density of 50 kg ha$^{-1}$, and retting with water. Young's modulus determination was not as accurate as the tensile

strength and density tests. Difficulties have been reported in the literature for Young's modulus determination, and fiber structure should be evaluated for a better understanding of Young's modulus behavior.

**Author Contributions:** Conceptualization, J.R. (João Ribeiro) and J.R. (João Rocha); Methodology, G.B. and J.R. (João Rocha); Software, G.B.; Validation, G.B.; Formal analysis, G.B. and M.R.M.; Investigation, J.R. (João Ribeiro), G.B. and M.R.M.; Resources, J.R. (João Ribeiro) and J.R. (João Rocha); Data curation, G.B. and M.R.M.; Writing—original draft, G.B.; Writing—review & editing, J.R. (João Ribeiro), M.R.M. and J.R. (João Rocha); Visualization, J.R. (João Rocha); Supervision, J.R. (João Ribeiro); Project administration, J.R. (João Ribeiro); Funding acquisition, J.R. (João Ribeiro) All authors have read and agreed to the published version of the manuscript.

**Funding:** Financial support was provided by Portugal's national funding FCT/MCTES (PIDDAC) to Centro de Investigação de Montanha (CIMO) (UIDB/00690/2020 and UIDP/00690/2020) and SusTEC (LA/P/0007/2020).

**Data Availability Statement:** Not applicable.

**Conflicts of Interest:** The authors declare no conflict of interest.

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
