# Peer review of "Experimental Study on Mechanical Properties of Hemp Fibers Influenced by Various Parameters"

_sustainability, doi:10.3390/su15129610_

Round 1

Reviewer 1 Report

Dear Authors,

Congratulations on your work, which is focused on a very interesting subject, and is properly organized.

In this work, the authors analyzed the effect of sowing density, nitrogen fertilization and extraction method on hemp fibers mechanical properties. The research appears to be efficiently done and appropriately reported, however, the standard of English must be improved and revised by a native English expert. Nevertheless, some questions and corrections must be answered to improve and complete the manuscript.

The reviewer has some comments detailed below and in the attached file.

1- The title of the manuscript is appropriate and concise.

2- The Abstract is properly organized, reflects the paper content, and summarizes the problem, and the conclusions.

3- Section 1 (Introduction): In this section, the authors do not indicate the novelty of their work. What is the innovation of the work when compared with the other researchers?

4- Section 2 (Materials and Methods): The authors could describe the experimental tests accomplished in their work in more detail.

5- Section 3 (Results): The results are not completely clear.

Were the results presented in Table 1 (for the Extraction Method, Sample Location, Seeding Density and Nitrogen Fertilization) obtained with fibers grown in Bragança or Mirandela?

In Figure 4 it seems that results obtained under the same conditions show different values (namely in the standard deviation; see an example marked in the attached file). This must be verified for the results presented in Figures 4 and 5.

6- Section 4 (Discussion): The discussion should be improved. For example, the authors could search for more bibliographic sources to compare with the results obtained.

The following sentence needs to be explained: “There is a linear trend line, well seen, for sowing densities of 25 and 50 Kg ha-1” (page 9, line 214). The trend line is the same for 25 and 50 Kg ha-1? Can you explain?

The following sentence needs to be explained: “A trend line can, also, be seen for sowing densities of 25 and 50 Kg ha-1, in figure 5” (page 9, line 218).

7- Conclusions. After a brief contextualization, the authors could use bullets to present the main conclusions.

8- Other comments:

8.1- Figure 1 should be improved.

8.2- Figure 2: Improve the legend description.

8.3- Figures 4 and 5: The legends of these figures need to be improved.

What is the meaning of the legends placed on the right side of these figures? What is the meaning of A25B, A50B, …….., N25B, N50B, ……., AM, NM? Please see the attached file.

8.4- Table 5: throughout the text the authors must always use the same number of decimal places.

8.5- Please always use a free space between units and values (example: 1N -> 1 N (page 4, line 124)).

8.6- Page 8, line 190: “average failure strength of 1.81%”; failure strength or elongation at break point (or failure strain)?

Good luck and best wishes.

Kind regards.

In general, the paper is well-written, organized and formatted. However, a light revision is recommended.

Author Response

The authors would like to acknowledge the careful revision done by the Reviewer, as well as to his/her precious suggestions and comments. Also, we would like to thank to the Editor for the opportunity that was given to us regarding the improvement of the original version of the manuscript (MS) and the further resubmission of the improved MS. Hence, we hope that this new version of the MS is worth publishing in the Journal Lubricants. All the changes are highlighted in yellow colour.

Congratulations on your work, which is focused on a very interesting subject, and is properly organized.

In this work, the authors analyzed the effect of sowing density, nitrogen fertilization and extraction method on hemp fibers mechanical properties. The research appears to be efficiently done and appropriately reported, however, the standard of English must be improved and revised by a native English expert. Nevertheless, some questions and corrections must be answered to improve and complete the manuscript.

The reviewer has some comments detailed below and in the attached file.

  • The title of the manuscript is appropriate and concise.

The authors would like to thank the reviewer for his kind comments.

  • The Abstract is properly organized, reflects the paper content, and summarizes the problem, and the conclusions.

The authors would like to thank the reviewer for his kind comments.

  • Section 1 (Introduction): In this section, the authors do not indicate the novelty of their work. What is the innovation of the work when compared with the other researchers?

The authors would like to thank the reviewer for his kind comments. The authors have expanded the last paragraph of section 1:

“In the present work, the contribution of this work is to analyze bundles of hemp fibers grown in Bragança and Mirandela regions, located in Portugal’s northeast for different conditions of Nitrogen fertilization and cultivation density. The main goal of this study was to determine the effects of Nitrogen fertilization, sowing density, ex-traction method, and sample location on hemp fibers’ mechanical properties with dif-ferent climatic conditions compared to those applied in other previous studies [16, 17].

  • Section 2 (Materials and Methods): The authors could describe the experimental tests accomplished in their work in more detail.

5- Section 3 (Results): The results are not completely clear.

Were the results presented in Table 1 (for the Extraction Method, Sample Location, Seeding Density and Nitrogen Fertilization) obtained with fibers grown in Bragança or Mirandela?

Yes. Since the geographic location is a characteristic of each fiber. In the statistical analysis, it was considered as two different groups to calculate the average of the density values. However, in the same table, other results of other different categories were shown. The table caption has been modified to highlight this issue.

In Figure 4 it seems that results obtained under the same conditions show different values (namely in the standard deviation; see an example marked in the attached file). This must be verified for the results presented in Figures 4 and 5.

Please note that the two analyses are independent based on two different groupings. In the first, tensile strength is with respect to the group (nitrogen fertilization), but in the second it is with respect to sowing density. If the values are similar is by chance. This has been clarified in this way:

“The average value for tensile strength, in all samples, was 565.34 ± 166.97 MPa with a specific tensile strength of 353.19 ± 114.02 MPa/g cm-3. Figure 4 shows the values of tensile strength with respect to the data grouped (Nitrogen fertilization and sowing density).”

6- Section 4 (Discussion): The discussion should be improved. For example, the authors could search for more bibliographic sources to compare with the results obtained.

The following sentence needs to be explained: “There is a linear trend line, well seen, for sowing densities of 25 and 50 Kg ha-1” (page 9, line 214). The trend line is the same for 25 and 50 Kg ha-1? Can you explain?

The following sentence needs to be explained: “A trend line can, also, be seen for sowing densities of 25 and 50 Kg ha-1, in figure 5” (page 9, line 218).

Thank you very much for your questions. We explained these questions in the manuscript:

  • After a brief contextualization, the authors could use bullets to present the main conclusions.

The authors thank the comment. However, we prefer to remain the conclusion in this form. We usually write the paper in this way.

8- Other comments:

8.1- Figure 1 should be improved.

The quality of Figure 1 has been improved.

8.2- Figure 2: Improve the legend description.

Thank you very much for your suggestion, the authors changed the legend.

8.3- Figures 4 and 5: The legends of these figures need to be improved.

What is the meaning of the legends placed on the right side of these figures? What is the meaning of A25B, A50B, …….., N25B, N50B, ……., AM, NM? Please see the attached file.

Thank you very much for your questions. We explained these questions in the manuscript:

8.4- Table 5: throughout the text the authors must always use the same number of decimal places.

8.5- Please always use a free space between units and values (example: 1N -> 1 N (page 4, line 124)).

The authors thank the comment. The misprint has been corrected.

8.6- Page 8, line 190: “average failure strength of 1.81%”; failure strength or elongation at break point (or failure strain)?

The authors thank the comment. The misprint has been corrected.

Good luck and best wishes.

Kind regards.

Reviewer 2 Report

Dear Author,

Address the comments given in the attachment.

Grammar check is needed for the journal article.

Author Response

The authors would like to acknowledge the careful revision done by the Reviewer, as well as to his/her precious suggestions and comments. Also, we would like to thank to the Editor for the opportunity that was given to us regarding the improvement of the original version of the manuscript (MS) and the further resubmission of the improved MS. Hence, we hope that this new version of the MS is worth publishing in the Journal Lubricants. All the changes are highlighted in yellow colour.

The paper is generally well written and structured. The findings of the paper are interesting to

the current research. This paper has a potential to be accepted, but some important points

have to be clarified.

General Comments

  1. The manuscript needs extensive revision for language and grammar.

The authors thank the comment. English has been checked through the paper.

  1. Symbols or units should be denoted correctly. For example, in many places, kilogram

for kilo it is written as Kg. It should be addressed throughout the manuscript by the

authors

The authors thank the comment. The misprint has been corrected.

Technical comments.

  1. Title need modification as : Influence of various parameters on mechanical properties of hemp fibers or Experimental study on mechanical properties of hemp fibers influenced by various parameters.

The authors thank the suggestions. We choose the second one “Experimental study on mechanical properties of hemp fibers influenced by various parameters”

  1. In the abstract, the first three lines need to be revised. (Line 9 to 11).

The authors thanks the comment. The sentence has been modified as follows:

“Hemp fibers produced by different extraction techniques which have been cultivated in the Por-tuguese districts of Bragança and Mirandela under various nitrogen fertilization and planting density conditions have been examined and mechanically described”

  1. Line 12 says optimizing the mechanical properties, but Line 232 and 237 says

optimal, reader is getting confused.

The authors thank for the comment. The sentence has been modified:

“The objective of this study was to assess the influence of nitrogen fertilization, sowing density, ex-traction method, and sample location in order to improve the mechanical properties of hemp fibers.”

  1. Section 2 – Line from 69 to 77 discuss about materials selection only. Physical properties and Chemical properties can be added in this section, if any

The authors thank for the comment. However, we did not find the physical and chemical properties.

  1. Figure 2 caption is very short , it can be more specific and detailed caption
  2. Section 3, for sub division 3.1 and 3.2 need more explanation, authors are just

reported the results and it can be referred with previous work it any.

The authors thank the comment. In the discussion section these results are commented and referenced with respect previous works.

  1. Figure 4 and 5 can be mentioned as Figure 4 (a) & (b) and caption can be mentioned respectively.

Thanks for the suggestion, but we prefer maintain like it is..

  1. Line 148, it is mentioned as Graphics, what exactly the word means. Authors need to

modify the word accordingly.

The authors thank the comment. The work has been removed.

  1. Separate sub section can be given for the Line 158 to 184 as ANOVA analysis.

The authors thank the comment. A new subsection 3.3. (ANOVA Analysis) has been created.

  1. Influence of parameter based on the experimental result need to explain.

Thank you for your suggestion. We included this information in the manuscript which are highlighted in yellow color.

  1. Conclusions the authors should concise the finding especially the novelty of the

paper.

The next sentence has been added:

“In this paper, the influence of different fiber parameters on mechanical properties has been studied. For this purpose, an experimental study has been carried out based on a design of experiments that has allowed the purchase of fibers with different characteristics and from different geographical areas of Portugal”.

Reviewer 3 Report

Dear Authors,

after reading the paper, I am submitting a review

Author Response

The authors would like to acknowledge the careful revision done by the Reviewer, as well as to his/her precious suggestions and comments. Also, we would like to thank to the Editor for the opportunity that was given to us regarding the improvement of the original version of the manuscript (MS) and the further resubmission of the improved MS. Hence, we hope that this new version of the MS is worth publishing in the Journal Lubricants. All the changes are highlighted in yellow colour.

Abstract:

The abstract is written correctly, all important information about the work is included.

Introduction:

Authors thanks the comment

The introduction is written correctly, addresses issues related to the topic of the paper. The

authors citations a lot of very good work.

The authors thank the comment

  1. Materials and Methods

The authors described in an easy-to-understand manner the section on obtaining fibers and the

methodology of the research conducted.

The authors thank the comment

Comments:

  • There int's description of analysis of variance (ANOVA).

The authors thank the comment. ANOVA has been explained in section 3.3:

“Contributions report the percentage that each source contributes to the total sequential sums of squares.”

“In order to evaluate the contribution of each parameter, a multivariate analysis based on ANOVA was, also, made for specific tensile strength measurements. The software used was Minitab.”

2) The Authors didn't mention in which statistical program they performed the analysis of

variance (ANOVA)

The software used was Minitab. It has been indicated: “The software used was  Minitab ”

Recommendations:

  • Please write about the ANOVA method in the (Materials and Methods) section.

ANOVA is mentioned in Materials and Methods section:

“The ANOVA evaluation was used to investigate which parameters were more influ-ential for Tensile Strength, Specific Tensile Strength, and Density of Hemp fibers.”

  • Please write in what program the statistical tests were performed.

It has been indicated in the previous response.

  1. Results

The authors present the results of fiber density and tensile strength tests. The tables and

figures are clear.

The authors thank the comment

Recommendations:

1) Dear authors, you present the results of variations in the form of tables. I believe that you

can show Pareto charts in addition.

Authors agree that Pareto is a very good tool but, since it is a multivariate analysis, Pareto charts will be very confusing and unintuitive.

  1. Discussion

The discussion of the results is detailed.

The authors thank the comment.

The authors compare the results of the study with the literature.

Analysis of variance (ANOVA) allowed the authors to predict which of the analyzed

parameters to influence on the improvement of the properties of the studied fibers.

I don't have any objections.

The authors thank the comment.

  1. Conclusion

Conclusions are formulated on the basis of the received studies.

I don't have any objections.

The authors thank the comment.

References

Literature selected for the topic of the research conducted.

21 items are cited.

I do not have any objections.

The authors thank the comment.

Evaluation of work:

In my opinion, the work is at a good scientific level, it was created as a result of a research

grant. Please include my comments in the work.

The authors thank the comment.